# RECURFORMER: NOT ALL TRANSFORMER HEADS NEED SELF-ATTENTION

## ABSTRACT

Transformer-based large language models (LLMs) excel in modeling complex language patterns but face significant computational costs during inference, especially with long inputs due to the attention mechanism's memory overhead. We observe that certain attention heads exhibit a distribution where the attention weights concentrate on tokens near the query token, termed as recency aware, which focuses on local and short-range dependencies. Leveraging this insight, we propose RecurFormer, a novel architecture that replaces these attention heads with linear recurrent neural networks (RNNs), specifically the Mamba architecture. This replacement reduces the cache size without evicting tokens, thus maintaining generation quality. RecurFormer retains the ability to model long-range dependencies through the remaining attention heads and allows for reusing pre-trained Transformer-based LLMs weights with continual training. Experiments demonstrate that RecurFormer matches the original model's performance while significantly enhancing inference efficiency. Our approach provides a practical solution to the computational challenges of Transformer-based LLMs inference, making it highly attractive for tasks involving long inputs.

## 1 INTRODUCTION

Transformer-based LLMs (OpenAI, 2023; Touvron et al., 2023) excel at modeling complex language patterns but come with significant computational costs. During inference, the prefill phase processes the input in parallel, generating the first token and initializing the key-value cache (KV-Cache), while the generation phase recursively produces each token, adding new keys and values to the KV-Cache (Radford et al., 2019; Brown et al., 2020). In long input tasks like document-based dialogue generation, the attention mechanism's overhead in the prefill phase leads to memory issues, making optimization crucial (Yang et al., 2024).

Currently, one of the optimization methods applicable to the prefill phase is PyramidInfer (Yang et al., 2024), which reduces the KV-Cache size by progressively removing tokens with lower attention weights. However, the intrinsic drawback is that token removal can decrease generation quality as the sequence length increases. Moreover, in modern Transformer-based inference frameworks, such as vLLM (Kwon et al., 2023), the prefix caching strategy is employed, which caches the KV-Cache from long documents or dialogue history to reduce computational complexity during the prefill phase in long document retrieval and multi-turn dialogue scenarios. Since it is difficult to predict which tokens may be needed in future token generation, such scenarios are not suitable for compression through token eviction.

Inspired by the dependency length minimization (DLM) phenomenon (Futrell et al., 2015) observed in quantitative linguistics and the principles of attention mechanisms, we identified a local, short-range token attention distribution pattern, referred to as the *recency aware*, where attention is concentrated on tokens close to the query token. Additionally, we discovered an attention distribution pattern where the attention weights are independent of relative position, termed *contextual retrieval*, characterized by attention being distributed across tokens at arbitrary positions in the sequence, reflecting a global focus.

While the attention mechanism excels at modeling dependencies over varying distances (Vaswani et al., 2017), we discover that **using a linear RNNs is a more efficient option for fitting the recency aware so that it reduces cache size requirements**. On one side, limited cache size makes linear

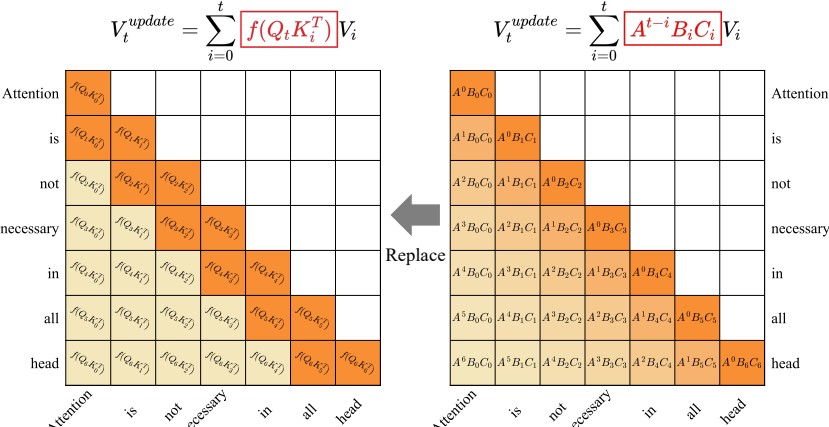

Figure 1: The left diagram shows how attention with recency aware updates values via weighted summation, where $f(x) = \text{softmax}(x/\sqrt{d_k})$, and $d_k$ is the dimension of key. The right diagram illustrates the linear RNNs update. $A$ represents the weights for state transitions, while $B_i$ and $C_i$ are input and output gates. Regions with darker shades of orange indicate a greater influence on $V_t^{\text{update}}$, such as representing higher attention weights, while lighter-colored regions have less influence.

RNNs less effective than Transformer in handling long-range dependencies (Jelassi et al., 2024; Arora et al., 2024), but it still captures local relationships effectively. On the other hand, linear RNNs have a lower cache size compared to the attention mechanism in both the parallel mode of the prefill phase and the recursive mode of the generation phase (Fu et al., 2023; Gu & Dao, 2023; Arora et al., 2024).

To avoid such inefficiency, we propose a novel structure named **RecurFormer**, which introduces linear **recur**rent structure to Trans**former** that achieves better efficiency for model inference. Specifically, it replaced the traditional attention mechanism with the Mamba architecture in attention heads exhibiting the recency aware, as shown in Figure 1, where Mamba is a linear RNNs based on selective structured state space sequence model, supporting parallel and recurrent computation (Gu & Dao, 2023). Compared to fully attention-based Transformer, RecurFormer reduces cache size in both the prefill and generation phases, without achieving these benefits by directly evicting any tokens. With continual training, RecurFormer can reuse the original model's weights while maintaining performance. The attention heads not replaced can still benefit from existing optimization strategies to further enhance overall performance. The advantages of RecurFormer include: 1) reducing cache size without evicting tokens, which helps maintain generation quality, and 2) the ability to reuse pre-existing Transformer-based model weights.

The key contributions of this paper are as follows:

- Inspired by the DLM phenomenon in quantitative linguistics and the computational principles of attention mechanisms, we are the first to observe that certain attention heads in Transformer-based LLMs can be effectively replaced by a Linear RNNs structure due to the recency aware property.
- We propose RecurFormer, a novel recursive-optimized Transformer architecture that replaces attention heads influenced by the recency aware property with the Mamba architecture, reducing cache size while reuse the original model weights.
- We show through HashHop experiments that RecurFormer matches the original model's quality. Continued training confirms performance recovery, and ablation studies with the multiple query associative recall (MQAR) task validate the need to retain certain attention mechanisms.

## 2 RELATED WORK

**Optimization Strategies for Efficient Inference.** Improving the computational and memory efficiency of Transformer-based LLMs has become a key research focus, driven by the need for scalable

inference in complex natural language processing applications. Optimization methods can be categorized into token eviction-based and non-eviction-based approaches. Token eviction-based strategies reduce storage and computation by dynamically removing less important tokens, functioning as a sparse attention mechanism. BigBird (Zaheer et al., 2020) pioneered this approach by retaining specific positional and random tokens, while H2O (Zhang et al., 2023) introduced attention-weight-based eviction to prioritize key information. Further optimizations, such as Scissorhands (Liu et al., 2024) and PyramidInfer (Yang et al., 2024), evict different tokens at each layer, with PyramidInfer optimizing the prefill phase by expelling tokens with low attention at each layer progressively. However, these methods become less efficient as the decoded sequence grows, especially in tasks requiring retention of dialogue history or external knowledge.

Non-eviction-based methods, such as DMC (Nawrot et al., 2024), maintain higher generation quality by merging the key and value of the current token with the last one in the KV-Cache. While effective at controlling KV-Cache growth, non-eviction methods face limitations in the prefill phase due to their sequential nature, which limits parallelization. This study aims to develop a non-eviction token optimization method applicable to both the prefill and generation phases, with the goal of reducing cache size in Transformer-based LLMs while maintaining generation quality comparable to the original model.

**Linear RNNs.** Non-linear RNNs, as the typical form of RNNs, is fundamentally constrained by non-linear dependencies in state transitions, which limit their ability to perform parallel computations across the sequence dimension. Inspired by physical systems, T-LSTM (Balduzzi & Ghifary, 2016) introduced a pioneering approach to linearize hidden state transitions by decoupling input and hidden states. This linearity enables cell states to be updated through associative operations. Building on this property, GILR-LSTM (Martin & Cundy, 2018) employs a parallel scan strategy to accelerate computation across sequence elements.

Recently, Linear RNNs based on state space models (SSM) have garnered considerable attention. By discretizing SSM, these models become applicable to discrete sequences, and when combined with the Hippo (Gu et al., 2020) memory update mechanism, they yield the structured state space (S4) model (Gu et al., 2022). Expanding upon S4, the selective structured state space (S6) model (Gu & Dao, 2023) further enhances sequence modeling capacity by increasing the dimensionality. By integrating S6 with gating architecture, the Mamba (Gu & Dao, 2023) was introduced, showing strong potential in sequence modeling tasks. However, due to limited memory capacity, Mamba still falls short in tasks such as sequence copying and long-range dependency modeling compared to attention-based architectures like Transformer (Jelassi et al., 2024). In RecurFormer, we identify scenarios that circumvent Mamba's limitations in modeling long-range dependencies, by focusing on cases where short-term dependencies are predominant.

## 3 METHODOLOGY

**Overview.** The process begins by identifying attention heads in the model that exhibit a strong short-range focus, characterized by the recency aware. Once identified, we replace the attention mechanism in these heads with a more efficient linear RNNs, which is better suited for modeling local dependencies. This replacement strategy allows us to maintain the model's generation ability while reducing cache size in both the prefill and generation phase. Our approach involves two steps:

- **Select replaced head**(Sec. 3.1): We first calculate the recency ratio (RR) for each attention head in the pre-trained LLMs and assign a recency aware index (RA-I). Attention mechanisms in heads with higher RA-I are prioritized for replacement by a linear RNNs.
- **Replace and continual training**(Sec. 3.2): We replace the attention mechanism in selected heads with a linear RNNs, enabling RecurFormer to capture local dependencies more efficiently. continual training is then conducted to restore performance, ensuring it matches the original Transformer while reducing computational costs during prefill and generation phases.

### 3.1 SELECTING HEADS FOR REPLACEMENT

The key insight behind RecurFormer is the observation that not all attention heads in a Transformer-based LLM exhibit the same focus distribution. We define the recency aware as the phenomenon

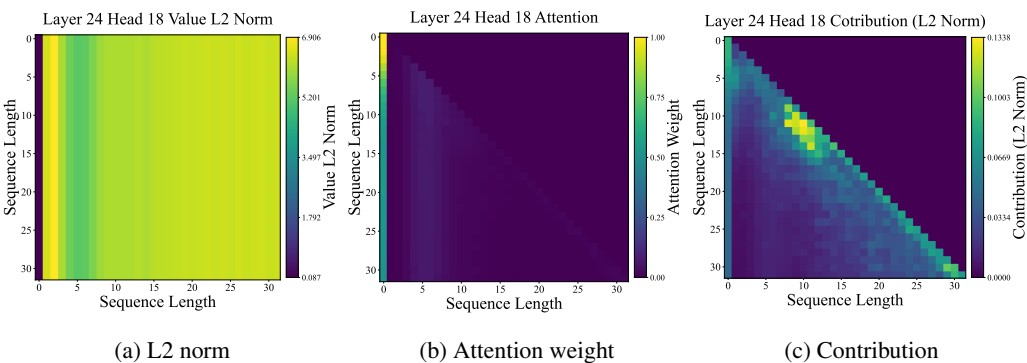

(a) L2 norm  (b) Attention weight  (c) Contribution

Figure 2: The figure presents averaged results from the 18th attention head of the 24th layer in the Llama-2-7b model, based on forward propagation through 1024 samples from Wikipedia English articles, each containing more than 1030 tokens, with the first 32 tokens from each sample selected as the input. The subfigures depict: (a) the L2 norm of the value vector for each token, (b) the attention weight matrix, and (c) the product of the L2 norm and the corresponding attention weight for each token, representing its contribution to the attention-weighted sum.

where the attention weights concentrate more on tokens that are temporally closer to the query token, while others exhibit a more uniform or distant focus across the sequence, which we call contextual retrieval. To quantify the degree of the recency aware in each head, we introduce the concept of RR, which is computed as

$$\text{RR}_h = \frac{\sum_{|i-j| \le k} A_{h,i,j}}{\sum_{i,j} A_{h,i,j}}, \tag{1}$$

where $A_{h,i,j}$ represents the attention weight from token $i$ to token $j$ in head $h$, and $k$ is a threshold that defines the local region of token interactions, i.e., the local range around the diagonal of the attention matrix. A high RR indicates that the head primarily focuses on recent tokens, suggesting that this head is mainly capturing short-range dependencies.

We begin by calculating the RR for each attention head to quantify its focus on recent tokens, aiming to identify attention distributions that align with the recency aware. For each sample, we compute the RR for every attention head and set a threshold $\alpha$. Attention heads with an RR exceeding $\alpha$ are recorded, and we tally the number of times each head is recorded across the dataset, defining this count as the head's RA-I.

However, during this process, we encountered a recurring issue where certain attention heads consistently assign disproportionately high attention weights to the first token. This attention sink phenomenon (Zaheer et al., 2020), as illustrated in Figure 2b, distorts the RR measurement. The excessive attention to the first token makes it difficult to accurately identify attention heads that are genuinely focused on short-range dependencies, which the recency aware is meant to capture.

To address this issue, we analyzed the characteristics of the first token. In causal attention mechanisms, the first token remains unaffected by subsequent tokens during attention-based information mixing, leading to a relatively fixed value distribution. When this special distribution is passed through the model's fully connected layers, it results in lower L1 and L2 norms compared to other tokens (Yan et al., 2024), as shown in Figure 2a. A token's contribution to the final output depends on both its attention weight and its value vector, which can be expressed as:

$$v_t^{\text{update}} = \sum_{i=0}^{t} A_{h,t,i} v_i, \tag{2}$$

where $v_t^{\text{update}}$ represents the updated value for token $t$, $A_{h,t,i}$ is the attention weight between token $t$ and token $i$ in head $h$, and $v_i$ is the value vector for token $i$. Despite often receiving higher attention weights, the first token's value vector exhibits lower L1 and L2 norms relative to other tokens, which limits its overall contribution to the final output (Guo et al., 2024; Devoto et al., 2024), as Figure 2c.

Table 1: Comparison of cache peak and cache size between self-attention with KV-cache and Mamba in the prefill and generation phases.

| Phase | Attention (KV-cache) | | Mamba | |
|---|---|---|---|---|
| | Cache Peak | Cache Size | Cache Peak | Cache Size |
| **Prefill** | $O(l_p^2 + l_p \cdot d)$ | $O(l_g \cdot d)$ | $O(l_p \cdot d)$ | $O(d)$ |
| **Generation** | $O(l_g \cdot d)$ | $O(l_g \cdot d)$ | $O(d)$ | $O(d)$ |

To mitigate the influence of the first token on RR calculations, we modify the RR calculation by excluding the first token. This adjustment provides a clearer measure of the head's focus on short-range dependencies, offering better insight into the recency aware.

Furthermore, when replacing the attention mechanism with a linear RNNs, the input gate in the linear RNNs provides an effective solution to this issue. It projects the value vector $v_t$ for each token in the sequence, adjusting the value distribution across all tokens so that the value vector of the first token can become similar to those of the other tokens. Specifically, this transformation is expressed as $v_i' = B_i v_i$, where $B_i$ is the weight vector of the input gate applied to $v_i$, the original value vector for token $i$. This projection allows the first token to appropriately influence other tokens, as it would in the attention mechanism, without requiring additional sequence-level modeling operations.

### 3.2 REPLACING SELECTED HEADS AND CONTINUAL TRAINING

Employing attention mechanisms in recency aware dominant heads results in inefficiencies. To address this, we propose substituting these heads with a linear RNNs structure, termed Mamba, within Transformer models. Mamba (Gu & Dao, 2023), based on the S6 architecture, is designed to model sequence information more efficiently. Although its performance in modeling long-term dependencies is constrained by finite memory compared to Transformer (Jelassi et al., 2024), it mitigates this limitation in scenarios where capturing local dependencies is critical. Additionally, since Mamba's state transitions do not rely on nonlinear activation functions, the operators within its state transitions are associative, allowing for efficient parallel computation via a parallel scan mechanism. This significantly enhances computational efficiency, particularly in the prefill phase, where parallel computation across multiple tokens is required.

Table 1 summarizes the cache peak and cache size of Mamba compared to self-attention with KV-cache, during both the prefill and generation phases. In this table, $l_p$ denotes the length of the input, $l_g$ the length of generated tokens, and $d$ the model dimension. As shown, Mamba reduces both the computational complexity and cache size across these phases.

In Mamba, the inference process for generating the next token requires two main cache states: the convolutional state and the SSM state. The convolutional state cache size and the SSM state cache size are given by

$$C_{\text{conv}} = d_{\text{inner}} \cdot d_{\text{conv}}, \quad C_{\text{SSM}} = d_{\text{inner}} \cdot d_{\text{state}}, \quad d_{\text{inner}} = k_{\text{epd}} \cdot d_{\text{model}}, \tag{3}$$

where $d_{\text{conv}}$ is the convolution kernel size, $d_{\text{state}}$ is the state dimension, $k_{\text{epd}}$ is expand factor, and $d_{\text{model}}$ is the model dimension. Compared to the cache size of Transformer, which increases with the sequence length during the generation phase, Mamba maintains a constant cache size during inference, effectively reducing memory pressure.

We denote the proportion of attention heads to be replaced as $\beta$, with those exhibiting higher RA-I being prioritized for replacement. In these selected heads, Mamba blocks substitute the attention mechanism, as described in Algorithm 1. $W_Q^{\text{part}}$ refers to the subset of weights from the original model's linear layer that is used to project the query from $X_{\text{in}}$, only including the necessary weights for the chosen attention heads, thus reducing unnecessary computations. Similarly, $W_K^{\text{part}}$ is used to project the key. In the S6 model, since each embedding dimension is processed independently, the inputs from all heads utilizing Mamba blocks are aggregated into a single tensor along the embedding dimension, allowing for efficient computation across multiple heads.

Since Mamba introduces new parameters, continual training RecurFormer is necessary to restore its performance to a level comparable to the original Transformer-based LLMs.

---

**Algorithm 1** RecurFormer Block

---

**Require:** $X_{\text{in}} \in \mathbb{R}^{B \times L \times D}$, list heads$_{\text{m}}$, list heads$_{\text{att}}$
**Ensure:** $X_{\text{out}} \in \mathbb{R}^{B \times L \times D}$
  1: $V \leftarrow W_V X_{\text{in}}$                    $\triangleright$ Value projections with weight matrix $W_V$
  2: $Q \leftarrow W_Q^{\text{part}} X_{\text{in}}$                    $\triangleright$ Partial query projections with $W_Q^{\text{part}}$
  3: $K \leftarrow W_K^{\text{part}} X_{\text{in}}$                    $\triangleright$ Partial key projections with $W_K^{\text{part}}$
  4: $X_{\text{att}} \leftarrow \text{AttentionBlock}(Q, K, V[\text{heads}_{\text{att}}])$              $\triangleright$ Multi-head self-attention output
  5: $X_{\text{m}} \leftarrow \text{MambaBlock}(V[\text{heads}_{\text{m}}])$                    $\triangleright$ Mamba block output
  6: $X_{\text{out}} \leftarrow \text{Concatenate}(X_{\text{att}}, X_{\text{m}})$   $\triangleright$ Concatenating outputs of attention heads and Mamba block

---

## 4 EXPERIMENTS

We evaluated RecurFormer on Qwen2 (Team, 2023) and Llama2 (Touvron et al., 2023) series to assess cache reduction and generation quality. Ablation studies explored different $\beta$ values, and continued training confirmed RecurFormer's ability to reuse pre-trained weights. We visualized the RA-I values of different heads across various models, alongside the attention distributions, and analyzed the contributions of the first token.

### 4.1 GENERATION QUALITY AND CACHE SIZE REDUCTION

**Backbones.**    We utilize the Qwen2-0.5B, Llama2-7B, and Qwen2-7B models as the base models, sourced from the official versions. Llama2-7B uses Multi-Head Attention (MHA), while Qwen2 series use Grouped Query Attention (GQA) (Ainslie et al., 2023).

**Task for Evaluating Generation Quality.**    Evaluating natural language generation quality is often subjective. To provide an objective measure, we use the linked list reasoning task, HashHop (Magic, 2024). In this task, the model reconstructs a linked list from its first element. We assess performance using the $h_{gq}$ metric, which is the ratio of the longest correct sequence starting from the first element to the total target list length, as shown in the $h_{gq}$ column of Table 2. This provides a clear measure of the model's ability to maintain quality as generation sequence length increases. Further details and examples are in Appendix A.1.

**Task for Statistic Cache Size.**    We evaluated Qwen2-0.5B, Llama2-7B, and Llama2-13B using randomly generated token sequences as inputs. The results, shown in the $cs_{10k}$ and $cs_{60k}$ columns of Table 2, demonstrate that RecurFormer not only increases the allowable input length but also significantly reduces cache size growth as the generation length extends. When $\beta$ is 0, corresponding to the original model, we define the cache size under this condition to be 1.0000.

**Experiment Implementation.**    RecurFormer was constructed based on the original models with $\beta$ set to 0.9, except for the model with 0.5B parameters, where $\beta$ was set to 0.5. All experiments were performed on a single A100 GPU with 80G of memory, with a batch size of 1. Further details on the computation of the RA-I and sample selection are provided in Appendix B.

**Main Results.**    The primary objective of our experiments was to demonstrate that RecurFormer can significantly reduce cache size while maintaining generation quality at a level comparable to unoptimized models. Specifically, we aimed to validate the effectiveness of RecurFormer across different attention mechanisms, such as MHA and GQA, as well as across models with varying parameter sizes. As shown in Table 2, RecurFormer achieved a cache size reduction of 89.7% at 10,240 tokens and 90.0% at 61,440 tokens for Llama2-7B, as indicated by the $cs_{10k}$ and $cs_{60k}$ columns, with only a minimal decrease in generation quality, where the $h_{gq}$ score is 0.839 compared to 0.894 for the unoptimized model. Similarly, for Qwen2-7B, RecurFormer reduced cache size by 87.3% at 10,240 tokens and by 87.5% at 61,440 tokens, confirming RecurFormer's ability to effectively manage cache size while preserving generation quality across different attention mechanisms, such as MHA and GQA.

Table 2: Performance comparison of RecurFormer across optimization methods for Qwen2-7B, Llama2-7B, and Qwen2-0.5B. The table shows the maximum input length ($l_m$), generation quality ($h_{gq}$), and cache size at generation lengths of 10,240 and 61,440 tokens.

| Model | Optimization Method | $l_m$ | $h_{gq}$ | $cs_{10k}$ | $cs_{60k}$ |
|---|---|---|---|---|---|
| **Qwen2-0.5B** | Original Model | 133,120 | 0.881 | 1.0000 | 1.0000 |
| | RecurFormer | 137,216 | 0.815 | 0.4390 | 0.4375 |
| **Llama2-7B** | Original Model | 71,680 | 0.894 | 1.0000 | 1.0000 |
| | RecurFormer | 91,800 | 0.839 | 0.1030 | 0.1000 |
| | PyramidInfer | 71,680 | 0.462 | 0.1054 | 0.1043 |
| **Qwen2-7B** | Original Model | 122,880 | 0.995 | 1.0000 | 1.0000 |
| | RecurFormer | 132,000 | 0.913 | 0.1268 | 0.1252 |

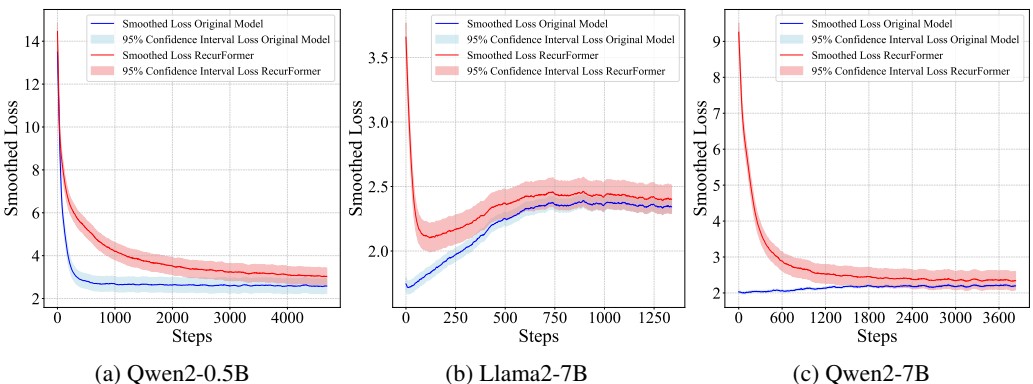

(a) Qwen2-0.5B     (b) Llama2-7B     (c) Qwen2-7B

Figure 3: Loss values of RecurFormer and the corresponding original models during continual training on the masked prediction task using the Wikipedia English training set.

Additionally, experiments on Qwen2-0.5B and Qwen2-7B further demonstrated the scalability of RecurFormer across models with different parameter sizes. As shown in Table 2, for Qwen2-0.5B, RecurFormer reduced cache size by 56.1% at 10,240 tokens and by 56.3% at 61,440 tokens, as seen in the $cs_{10k}$ and $cs_{60k}$ columns, while maintaining generation quality comparable to the unoptimized model, as indicated by the $h_{gq}$ column. This consistency in simultaneously preserving generation quality and optimizing cache size and memory requirements across models with different parameter sizes highlights the robustness of RecurFormer. Furthermore, RecurFormer increased the maximum input length for all tested models, extending the limit for Llama2-7B from 71,680 tokens to 91,800 tokens and for Qwen2-7B from 122,880 tokens to 132,000 tokens, as shown in the $l_m$ column, demonstrating its potential for long-sequence generation tasks.

In the case of Llama2-7B, both RecurFormer and PyramidInfer achieved similar cache size reductions. RecurFormer reduced the cache size by 89.7% at 10,240 tokens and 90.0% at 61,440 tokens, while PyramidInfer reduced it by 89.46% and 89.57%, respectively. However, the key difference lies in the generation quality. RecurFormer maintained a high generation quality with an $h_{gq}$ score of 0.839, only slightly lower than the unoptimized model's 0.894. In contrast, PyramidInfer's $h_{gq}$ score dropped significantly to 0.462, indicating a substantial degradation in generation quality compared to the original model. This highlights RecurFormer's ability to balance cache optimization with minimal impact on generation quality, a clear advantage over PyramidInfer.

## 4.2 CONTINUAL TRAINING

To validate the effectiveness of RecurFormer in inheriting the weights from Transformer-based LLMs, we performed continued training on RecurFormer with a $\beta$ value of 0.5, constructed from Qwen2-0.5B, Llama2-7B, and Qwen2-7B. The training was conducted using a masked prediction task on the Wikipedia English training set. The results, presented in Figure 3, demonstrate RecurFormer's ability to converge with the original models and achieve comparable performance.

Table 3: PPL values for the original models and RecurFormer on the masked prediction task using the Wikipedia English validation set.

| Model | Qwen2-0.5B | | Llama2-7B | | Qwen2-7B | |
|---|---|---|---|---|---|---|
| | Original | RecurFormer | Original | RecurFormer | Original | RecurFormer |
| PPL | 12.088 | 14.202 | 9.023 | 9.199 | 9.215 | 9.692 |

Table 4: Expansion of RecurFormer across different models with various $\beta$ values.

| Model | Qwen2-0.5B | | | | | | Llama2-7B | | | | | | Qwen2-7B | | | | | |
|---|---|---|---|---|---|---|---|---|---|---|---|---|---|---|---|---|---|---|
| $\beta$ | 0.00 | 0.25 | 0.50 | 0.75 | 0.90 | 1.00 | 0.00 | 0.25 | 0.50 | 0.75 | 0.90 | 1.00 | 0.00 | 0.25 | 0.50 | 0.75 | 0.90 | 1.00 |
| $l_{max}$ | 133,120 | 135,168 | 137,216 | 139,264 | 142,048 | 143,360 | 71,680 | 75,080 | 78,480 | 81,920 | 91,800 | 94,208 | 122,880 | 125,440 | 128,000 | 130,560 | 132,000 | 133,120 |
| $h_{gq}$ | 0.881 | 0.846 | 0.815 | 0.051 | 0.000 | 0.000 | 0.894 | 0.849 | 0.861 | 0.818 | 0.839 | 0.000 | 0.995 | 0.987 | 0.986 | 0.922 | 0.913 | 0.000 |
| $cs_{10k}$ | 1.0000 | 0.7043 | 0.4390 | 0.3145 | 0.0632 | 0.0006 | 1.0000 | 0.7510 | 0.5020 | 0.2530 | 0.1030 | 0.0040 | 1.0000 | 0.6578 | 0.3815 | 0.5930 | 0.1268 | 0.0007 |
| $cs_{60k}$ | 1.0000 | 0.7035 | 0.4375 | 0.3130 | 0.0626 | 0.0001 | 1.0000 | 0.7500 | 0.5000 | 0.2500 | 0.1000 | 0.0010 | 1.0000 | 0.6563 | 0.3800 | 0.5900 | 0.1252 | 0.0001 |

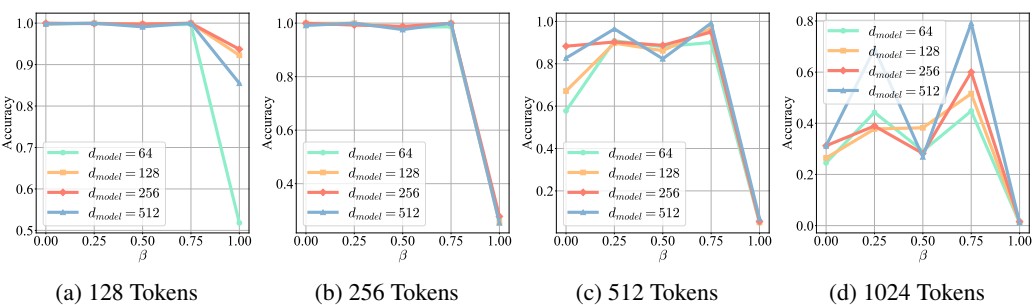

(a) 128 Tokens    (b) 256 Tokens    (c) 512 Tokens    (d) 1024 Tokens

Figure 4: Accuracy in the MQAR task for different sequence lengths. Each subfigure shows the accuracy of RecurFormer and the original Transformer model over validation samples of (a) 128 tokens, (b) 256 tokens, (c) 512 tokens, and (d) 1024 tokens.

After convergence, we evaluated both RecurFormer and the corresponding original models on the validation set, using the masked prediction task and calculating the perplexity (PPL), as Table 3.

### 4.3 ABLATION STUDIES

To verify the necessity of retaining attention heads in RecurFormer, we extended Table 2 with different $\beta$ values. When $\beta$ is set to 1.0, meaning no attention heads are retained in RecurFormer, we observe a significant degradation in generation quality in Table 4, with $hp_{gq}$ dropping to 0. This phenomenon occurs consistently across models with different parameter sizes, as well as those utilizing either MHA or GQA as the base models. Intrigued by this finding, we conducted additional experiments using the MQAR task on randomly initialized Transformer and RecurFormer models.

The MQAR task requires the model to return the value corresponding to a given key in a sequence of key-value pairs, with a detailed explanation of the task provided in Appendix A.2. The training set consists of samples with 4 to 64 pairs, and pair lengths ranging from 64 to 256. The test set contains samples with 4 to 256 pairs, and lengths ranging from 64 to 1,024. We used a randomly initialized Transformer model with rotary position encoding (RoPE) (Su et al., 2024) and GQA, specifically with 2 layers, 512 embedding dimensions, 8 attention heads, and 4 key-value heads. After selecting the heads to be replaced in each layer based on $\beta$, we trained the model on the training set.

We recorded the proportion of correct value predictions, i.e., accuracy, on the validation set for different $\beta$ values, as shown in Figure 4. Specific examples and descriptions of the MQAR task are provided in Appendix A.2. Despite the MQAR task being simpler than the HashHop task, we observed that RecurFormer, with all attention heads replaced by Mamba blocks, still exhibited poor accuracy, indicating that retaining the attention mechanism in RecurFormer is necessary.

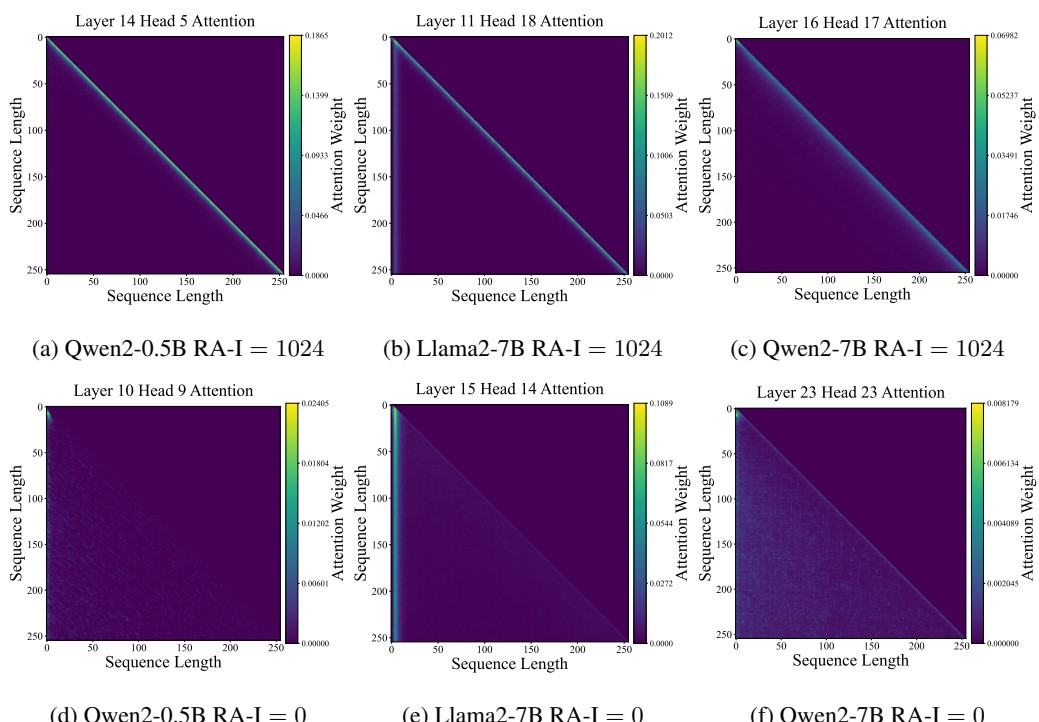

(a) Qwen2-0.5B RA-I $= 1024$     (b) Llama2-7B RA-I $= 1024$     (c) Qwen2-7B RA-I $= 1024$

(d) Qwen2-0.5B RA-I $= 0$     (e) Llama2-7B RA-I $= 0$     (f) Qwen2-7B RA-I $= 0$

Figure 5: Visualization of attention distribution for heads with high, mid, and low RA-I values in the Qwen2-0.5B, Llama2-7B, and Qwen2-7B models, averaged over 1024 samples. High RA-I heads focus more on recent tokens (main diagonal), while low RA-I heads show a more global attention pattern, with notably higher attention on the first token.

## 4.4 ANALYTICAL AND STATISTICAL EXPERIMENTS

**RA-I Visualization** We selected attention heads with different RA-I values from the Qwen2-0.5B, Llama2-7B, and Qwen2-7B models and visualized their behavior, as shown in Figure 5. The figure presents the average attention distribution heatmaps for these heads over 512 English samples, each containing more than 1030 tokens. For the analysis, we used the first 256 tokens of each sample as input. Attention heads with a high RA-I tend to focus on tokens close to the query token, resulting in a bright main diagonal that reflects the model's emphasis on local context dependencies. In contrast, heads with a low RA-I exhibit a more uniform attention distribution across the entire sequence, capturing longer-range dependencies. The attention distribution for low RA-I heads displays a mosaic-like pattern, indicating a more global focus. This distinction suggests that high RA-I heads are better suited for handling short-range dependencies, while low RA-I heads play a critical role in capturing long-range information. Although we did not include the first token in the visualization, it was still part of the attention calculation.

Additionally, in Figure 6, we visualized the RA-I values of each head in each layer of the Qwen2-0.5B, Llama2-7B, and Qwen2-7B models. It can be observed that more heads in the Qwen2 series exhibit higher RA-I values, indicating that the heads in the Qwen2 models more frequently display a recency aware attention distribution pattern.

**Token Contribution Analysis.** To verify the hypothesis that the contribution of the first token's value vector is lower than that of other tokens, we analyzed the contribution values across different heads in the Qwen2-0.5B, Llama2-7B, and Qwen2-7B models. The contribution value is computed as the product of the attention weight and the L2 norm of the value vector, as discussed in Section 3.1. We used 512 samples from the Wikipedia English dataset, each containing more than 1030 tokens. From these, the first 256 tokens were selected for analysis. Table 5 presents the mean contributions along with their 95% confidence intervals for both the first token and non-first tokens.

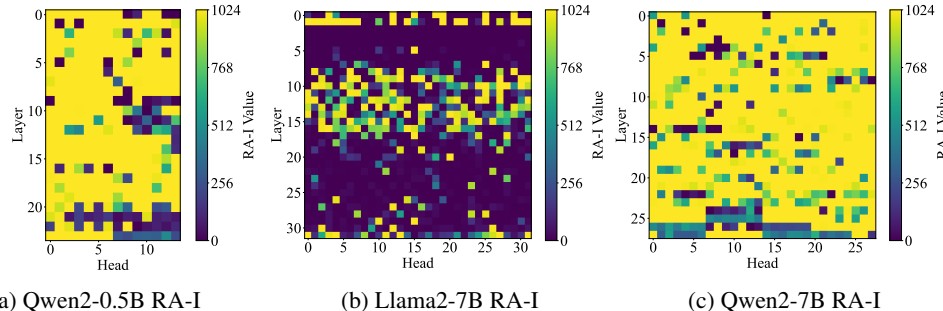

(a) Qwen2-0.5B RA-I  (b) Llama2-7B RA-I  (c) Qwen2-7B RA-I

Figure 6: RA-I values of each head in each layer of the Qwen2-0.5B, Llama2-7B, and Qwen2-7B models. We observed generally higher RA-I values in the Qwen2 model series, suggesting that more attention heads are primarily recency aware.

Table 5: Mean L1 and L2 Contribution Values and 95% Confidence Intervals (CI) for First and Non-First Tokens in Qwen2-0.5B, Llama2-7B, and Qwen2-7B.

| Model | L1 Contribution | | L2 Contribution | |
|---|---|---|---|---|
| | Mean | 95% CI | Mean | 95% CI |
| Qwen2-0.5B | 0.0281 | [-0.0530, 0.1094] | 0.0070 | [-0.0083, 0.0222] |
| | 0.0718 | [0.0286, 0.1152] | 0.0129 | [0.0046, 0.0212] |
| Llama2-7B | 0.0082 | [-0.0320, 0.0486] | 0.0013 | [-0.0041, 0.0067] |
| | 0.0442 | [0.0254, 0.0630] | 0.0052 | [0.0030, 0.0074] |
| Qwen2-7B | 0.0708 | [-0.5195, 0.6602] | 0.0120 | [-0.0713, 0.0947] |
| | 0.1465 | [0.0615, 0.2314] | 0.0181 | [0.0075, 0.0286] |

As shown in Table 5, the mean contribution of the first token is significantly lower than that of non-first tokens in the Qwen2-0.5B and Llama2-7B models, as indicated by the non-overlapping confidence intervals. Although the variability in the first token's contribution is higher in the Qwen2-7B model, the trend remains consistent, with the first token contributing less than non-first tokens. These results support the hypothesis that the first token's contribution is limited, providing a basis for ignoring the first token when calculating the RR in practical scenarios.

## 5  CONCLUSION

We explored the necessity of self-attention in all heads of Transformer-based LLMs and demonstrated the potential of replacing some self-attention mechanisms with a Linear RNNs. By identifying the recency aware phenomenon, where certain attention heads focus on recent tokens, we introduced RecurFormer. In RecurFormer, we replace self-attention with a linear recurrent neural network in these specific heads. This substitution reduces computational costs and cache size during both the prefill and generation phases without compromising performance. Our experiments showed that RecurFormer effectively reuses pretrained Transformer weights with continual training, maintaining generation quality while enhancing efficiency. However, RecurFormer faces a key limitation in efficiently parallelizing the computation of Mamba blocks and self-attention heads within the same layer. This challenge is particularly evident under small batch sizes, affecting hardware resource utilization, which we aim to improve in future work. By integrating linear recurrence into Transformer architectures, RecurFormer provides a promising foundation for future large-scale model architectures, potentially leading to more efficient and scalable neural networks.

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

## A    TASKS DESCRIPTIONS

### A.1    HASHHOP

HashHop provides an objective evaluation of how the quality of generated content changes with sequence length. In this task, the model must sequentially reconstruct a linked list, starting from the first element. Each element in the linked list is represented as a random string of length $h_e$, with the target list containing $h_p$ connections, and the total input consisting of $h_l$ characters.

We use the $h_{gq}$ metric to evaluate the model's performance, defined as the ratio of the longest correct sequence starting from the first element to the total length of the target list. This metric objectively reflects the model's ability to preserve generation quality as the sequence grows.

For example:

$$10 \rightarrow 17, \quad 04 \rightarrow 05, \quad 01 \rightarrow 02, \quad 62 \rightarrow 23, \quad 02 \rightarrow 03,$$
$$99 \rightarrow 85, \quad 21 \rightarrow 34, \quad 03 \rightarrow 04, \quad 42 \rightarrow 73, \quad 05 \rightarrow 06.$$

In this case, the valid linked list is:

$$01 \Rightarrow 02 \rightarrow 03 \rightarrow 04 \rightarrow 05 \rightarrow 06.$$

Here, the elements in light gray represent random pairs that serve as distractors, and the symbol $\Rightarrow$ indicates the starting point of the linked list. The task challenges the model to connect pairs where the second element of one pair matches the first of another, forming a valid linked list.

### A.2    MQAR

For example:

$$A \rightarrow 4, \quad F \rightarrow 1, \quad B \rightarrow 3, \quad C \rightarrow 6$$

For the queries:

$$A \,?, \quad C \,?, \quad F \,? \quad \Rightarrow \quad 4, \quad 6, \quad 1.$$

Here, the ? indicates the position where the model needs to predict the corresponding value for each key based on the provided key-value pairs.

## B    EXPERIMENT IMPLEMENTATION DETAIL

**Calculate RA-I.**    From the Wikipedia English dataset, we randomly selected 1024 samples where the number of tokens exceeded 1030. For each sample, we extracted the first 1024 tokens to compute the RA-I for each attention head in the Transformer-based LLMs, setting $\alpha$ to 102.

**Configuration of Mamba block.**    The Mamba block parameters were configured as follows: $d_{\text{conv}}$ was set to 4, $d_{\text{state}}$ was set to 16, $dt_{\text{rank}}$ was set to 256, and $k_{\text{epd}}$ was set to 2.

**Configuration of HashHop task.**    The HashHop dataset was generated with $h_e$ set to 8, $h_p$ set to 16, and $h_l$ set to 6144. We continued training RecurFormer on the HashHop dataset, experimenting with various values of $\beta$. After completing the continued training on the HashHop dataset, the model's performance was evaluated using the $h_{gq}$ metric.

**Configuration of PyramidInfer.**    We configured PyramidInfer by adjusting the inter-layer token eviction ratio $P_{\text{reduce}}$ and the minimum token retention count $P_{\text{min}}$ to match the cache size as closely as possible with our approach. For Llama2-7B it was set to 0.7, and $P_{\text{min}}$ was set to 32.

