# OpenReview forum: "RecurFormer: Not All Transformer Heads Need Self-Attention"
_ICLR.cc/2025/Conference — ICLR 2025 Conference Withdrawn Submission_

### Official Review · Reviewer_7Nho · 2024-10-30

**Soundness:** 2
**Presentation:** 2
**Contribution:** 2
**Rating:** 5
**Confidence:** 4

**Summary:**

The authors introduce a cache compression technique that identifies "recency-aware" (RA) heads, and then proposes to replace them with linear RNN variant (in this case, Mamba), with the goal of minimal performance loss after continual training.

First, given a pre-trained Transformer, the RA heads are identified by looking at the "recency ratio", the heads where attention weighs temporally close to the query token. Then a threshold is decided and all heads with a recency ratio above it are deemed to be RA; continued pre-training (or up-training) tokens replaces attention in these heads with Mamba.

Qwen2 and Llama2 variants are up-trained with this technique, and large cache size reductions are demonstrated, while PPL for the original vs RecurFormer variant is computed on Wikipedia English val set, and ablations for MQAR are demonstrated.

**Strengths:**

The motivation behind the paper is solid – cache reduction through token eviction is problematic, and the “linearization” of recency-aware heads is an intriguing and novel approach for non-eviction based cache reduction.

**Weaknesses:**

The major issue is the lack of experimental validation, which is quite limited.  Standard language benchmarks (Hellaswag, MMLU, etc.) that the base models were evaluated on is missing, so it is not possible to gauge the extent to which the proposed approaches degrades (or doesn’t degrade) NLU performance.

Though the MQAR ablations are suggestive of possible strengths relative to pure linear models for longer-context tasks, prior work has shown that linearized models [3] struggle at long context tasks like SCROLLS [4], so degradations on standard NLU long-context evaluations should also be investigated.

Also missing are some references to prior work that takes pre-trained vanilla Transformers and up-trains some or all blocks into efficient alternatives, for the encoder-decoder [1], BERT [2] and LLM [3] setting.

[1] Kasai, Jungo, Hao Peng, Yizhe Zhang, Dani Yogatama, Gabriel Ilharco, Nikolaos Pappas, Yi Mao, Weizhu Chen, and Noah A. Smith. "Finetuning pretrained transformers into rnns." EMNLP 2021

[2] Zhang, Michael, Kush Bhatia, Hermann Kumbong, and Christopher Ré. "The hedgehog & the porcupine: Expressive linear attentions with softmax mimicry." ICLR 2024

[3] Mercat, Jean, Igor Vasiljevic, Sedrick Keh, Kushal Arora, Achal Dave, Adrien Gaidon, and Thomas Kollar. "Linearizing Large Language Models." COLM 2024

[4] Shaham, Uri, et al. "Scrolls: Standardized comparison over long language sequences." EMNLP 2022

**Questions:**

How many tokens are needed for the conversion fine-tuning / continual training? It's unclear exactly what the continual training protocol is in the experimental validation; have the authors trained their model on standard language pre-training data (e.g. redpajama)?

I think the major missing piece in the evaluation would be to run the fine-tuned Qwen and LLaMA models on the standard NLU evaluation harnesses for both standard (e.g. Hellaswag, MMLU) and long-context (e.g. Qasper and NarrativeQA in SCROLLS). Showing minimal regression on these tasks would go a long way to validating the approach, but also gaps on some benchmarks may be illustrative of where the "recency-aware" heads may be more or less important, which would be an interesting finding.

---

### Official Review · Reviewer_6T8B · 2024-11-03

**Soundness:** 1
**Presentation:** 3
**Contribution:** 2
**Rating:** 3
**Confidence:** 4

**Summary:**

This paper presents RecurFormer, an innovative architecture designed to reduce the computational demands of Transformer-based LLMs during long-context  inference. By observing that certain attention heads in LLMs predominantly focus on nearby tokens, the paper proposes replacing these heads with Mamba architecture. This substitution minimizes memory overhead without sacrificing token information, thus maintaining generation quality.

**Strengths:**

1. The idea presented in this paper of converting short-sighted attention heads into linear attention has innovation.
2. The presentation is clear.

**Weaknesses:**

1. The main issue with this paper is that the experiments are not comprehensive enough, as they are only conducted on two synthetic datasets. Additional experiments on a wider range of synthetic datasets and real-world tasks, such as InfiniteBench and LongBench, are needed.
2. The baselines used for comparison are insufficient. Some straightforward solutions, such as converting short-sighted attention heads into sliding window attention like Razor Attention [1], or using linear attention to focus on long-term tokens like LESS [2], should also be included for comparison.
3. The comparison with PyramidInfer is unfair, as PyramidInfer is a training-free method. For a fair comparison, PyramidInfer should be integrated into the model architecture during training.

[1] https://arxiv.org/pdf/2407.15891
[2] https://arxiv.org/pdf/2402.09398

**Questions:**

See weaknesses above.

---

### Official Review · Reviewer_Rp5i · 2024-11-03

**Soundness:** 3
**Presentation:** 3
**Contribution:** 3
**Rating:** 3
**Confidence:** 4

**Summary:**

This paper proposes a non-eviction token optimization method that selectively replaces attention heads influenced by the recency-aware property with the Mamba architecture, reducing cache size while reusing part of the original model weights. RecurFormer maintains the original model’s performance on certain tasks, with enhanced inference efficiency.

**Strengths:**

The paper introduces RecurFormer, a practical solution that integrates linear RNNs into Transformer models, specifically replacing certain attention heads with the Mamba architecture to enhance inference efficiency. This approach effectively reduces inference cache size without evicting tokens, thereby maintaining generation quality while addressing the computational challenges associated with Transformer-based LLMs. The methodology is well-documented, provide a detailed overview of the selection and replacement process, including the calculation of the recency ratio and the use of continual training to restore performance.

**Weaknesses:**

1. In Table 3, the loss comparison indeed shows a noticeable PPL gap between models of different scales compared to the original structure, particularly for the 0.5B parameter model where β was set to 0.5, yet the PPL still increased by more than 2.

2. The paper lacks a performance comparison with MQA, as well as comparisons of hybrid approaches between different heads and layers. This omission makes it difficult to assess RecurFormer's relative advantages in these areas.

3. The benchmarks used for validation are relatively narrow, focusing mainly on HashHop and MQAR tasks. There is a lack of evaluation on the model's general capabilities using common academic benchmarks like HellaSwag, MMLU, and C-Eval, as well as other long-sequence evaluation tasks such as LongBench and LongEval.

4. The method is practical and effective to some extent, but the starting point—different heads focusing on different ranges—and the implemented methods lack innovation.

**Questions:**

1. Is there a comparison with MQA, eviction-based methods, and hybrid approaches between layers? This would be crucial for a comprehensive evaluation of RecurFormer's performance.

2. Besides HashHop and MQAR, are there evaluations on common academic benchmarks like HellaSwag, MMLU, and C-Eval, as well as other long-sequence evaluation tasks such as LongBench and LongEval?

3. Will replacing part of the heads with RNN (Mamba) introduce new parameters? If so, how does the parameter count compare with the baseline models?

---

### Official Review · Reviewer_3o57 · 2024-11-03

**Soundness:** 2
**Presentation:** 2
**Contribution:** 2
**Rating:** 3
**Confidence:** 4

**Summary:**

This paper proposed RecurFormer, which replaces parts of attention heads with linear recurrent neural networks (RNNs) to focus on local dependencies, reducing memory usage. RecurFormer retains long-range dependency modeling and allows for the reuse of pre-trained weights. Experiments tries to show it matches the original model's performance while enhancing inference efficiency.

**Strengths:**

This paper is relatively easy to understand, presentation of the method is clear. And the idea of replacing parts of the attention heads with Mamba instead of all is worth exploring.

**Weaknesses:**

However, the experimental results are pretty weak.
- The only evaluation metric used is HashHop, which gives very limited idea of the finetuned model's performance. The author should consider evaluate on more general and standard langauge tasks, such as the ones in lm-evaluation-harness.
- To evaluate performance on the long context reasoning ability, the author should consider standard eval on passkey retrival dataset.
- The proposed approach can be considered as within-layer hybrid mamba. It would interesting to compare the performance with between-layer hybrid Mamba, such as Jamba, Lieber et al., 2024, and Samba, Ren et al., 2024.

**Questions:**

- I wonder what is performance difference between sliding window attention and proposed method when using similar amount of compute.
- more eval results as mentioned in the weakness section.

---

### Note · Authors · 2024-11-29

**Comment:**

We are very sorry that we were unable to complete the additional experiments suggested by the reviewers before the deadline due to the sheer volume of experiments.

We thank the reviewers for their labor and we will take your suggestions to improve our paper.

**Withdrawal Confirmation:**

I have read and agree with the venue's withdrawal policy on behalf of myself and my co-authors.